# PROVABLE KNOWLEDGE TRANSFER USING SUCCESSOR FEATURE FOR DEEP REINFORCEMENT LEARNING

## ABSTRACT

This paper studies the transfer reinforcement learning (RL) problem where multiple RL problems have different reward functions but share the same underlying transition dynamics. In this setting, the Q-function of each RL problem (a.k.a. a task) can be decomposed into a successor feature (SF) and a reward mapping: the former characterizes the transition dynamics, and the latter characterizes the task-specific reward function. This Q-function decomposition, coupled with a policy improvement operator known as generalized policy improvement (GPI), reduces the search space of finding the optimal Q-function, and the SF & GPI framework exhibits promising empirical performance compared to traditional RL methods like Q-learning. However, its theoretical foundations remain largely unestablished, especially when learning successor features using deep neural networks (SFs-DQN). This paper studies the provable knowledge transfer using SFs-DQN in transfer RL problems. We establish the first convergence analysis with provable generalization guarantees for SF-DQN with GPI. The theory reveals that SF-DQN with GPI outperforms conventional RL approaches, such as deep Q-network, in terms of both faster convergence rate and better generalization. Numerical experiments on real and synthetic RL tasks support the superior performance of SF-DQN & GPI, quantitatively aligning with our theoretical findings.

## 1 INTRODUCTION

In reinforcement learning (RL), the goal is to train an agent to perform a task within an environment in a desirable manner by allowing the agent to interact with the environment. Here, the agent is guided towards the desirable behavior by the rewards, and the optimal policy is derived from a learned value function (Q-function) in selecting the best actions to maximize the immediate and future rewards. This framework can effectively capture a wide array of real-world applications, such as gaming(Mnih et al., 2013; Silver et al., 2017), robotics (Kalashnikov et al., 2018), autonomous vehicles (Shalev-Shwartz et al., 2016; Schwarting et al., 2018), healthcare (Coronato et al., 2020), and natural language processing (Tenney et al., 2018). However, RL agents require a significant amount of interactions with the environment to tackle complex tasks, especially when RL is equipped with deep neural networks (DNNs). For example, AlphaGo (Silver et al., 2017) required 29 million matches and 5000 TPUs at a cost exceeding \$35 million, which is time-consuming and memory-intensive. Nevertheless, many complex real-world problems can naturally decompose into multiple interrelated sub-problems, all sharing the same environmental dynamics (Sutton et al., 1999; Bacon et al., 2017; Kulkarni et al., 2016a). In such scenarios, it becomes highly advantageous for an agent to harness knowledge acquired from previous tasks to enhance its performance in tackling new but related challenges. This practice of leveraging knowledge from one task to improve performance in others is known as transfer learning (Lazaric, 2012; Taylor & Stone, 2009; Barreto et al., 2017).

This paper focuses on an RL setting with learning multiple tasks, where each task is associated with a different reward function but shares the same environment. This setting naturally arises in many real-world applications such as robotics (Yu et al., 2020). We consider exploring the knowledge transfer among multiple tasks via the successor feature (SF) framework (Barreto et al., 2017) which disentangles the environment dynamic from the reward function at an incremental computational cost. The SF framework is derived from successor representation (SR) (Dayan, 1993) by introducing the value function approximation. Specifically, SR (Dayan, 1993) decouples the value function into a future state occupancy measure and a reward mapping. Here, the future state occupancy

characterizes the transition dynamics of the environment, and the reward mapping characterizes the reward function of the task. SF is a natural application of SR in solving value function approximation. Furthermore, Barreto et al. (2017) propose a generalization of the classic policy improvement, termed generalized policy improvement (GPI), enabling smooth knowledge transfer across learned policies. In contrast to traditional policy improvement, which typically considers only a single policy, Generalized Policy Improvement (GPI) operates by maintaining a set of policies, each associated with a distinct skill the agent has acquired. This approach enables the agent to switch among these policies based on the current state or task requirements, providing a flexible and adaptive framework for decision-making. Empirical findings presented in (Barreto et al., 2017) highlight the superior transfer performance of SF & GPI in deep RL when compared to conventional methods like Deep Q-Networks (DQNs). Subsequent works further justified the improved performance of SF in subgoal identification (Kulkarni et al., 2016b) and real-world robot navigation (Zhang et al., 2017).

While performance guarantees of SF-based learning are provided in the simple tabular setting (Barreto et al., 2017; 2018), less is known for such approaches in the widely used function approximation setting. In this context, this paper aims to close this gap by providing theoretical guarantees for SF learning in the context of DNNs. Our objective is to explore the convergence and generalization analysis of SF when paired with DNN approximation. We also seek to delineate the conditions under which SF learning can offer more effective knowledge transfer among tasks when contrasted with classical deep reinforcement learning (DRL) approaches, e.g., DQN (Mnih et al., 2013).

**Contributions.** This paper presents the first convergence analysis with generalization guarantees for successor feature learning with deep neural network approximation (SF-DQN). This paper focuses on estimating the optimal Q-value function through the successor feature decomposition, where the successor feature decomposition component is approximated through a deep neural network. The paper offers a comprehensive analysis of the convergence of deep Q-networks with successor feature decomposition and provides insights into the improved performance of the learned Q-value function derived from successor feature decomposition. The key contributions of this study are as follows:

**C1. The convergence analysis of the proposed SF-DQN to the optimal Q-function with generalization guarantees.** By decomposing the reward into a linear combination of the transition feature and reward mapping, we demonstrate that the optimal Q-function can be learned by alternately updating the reward mapping and the successor feature using the collected data in online RL. This learned Q-function converges to the optimal Q-function with generalization guarantees at a rate of $1/T$, where $T$ is the number of iterations in updating transition features and reward mappings.

**C2. The theoretical characterization of enhanced performance by leveraging knowledge from previous tasks through GPI.** This paper characterizes the convergence rate with generalization guarantees in transfer RL utilizing GPI. The convergence rate accelerates following the degree of correlation between the source and target tasks.

**C3. The theoretical characterization of the superior transfer learning performance with SF-DQN over non-representation learning approach DQNs.** This paper quantifies the transfer learning ability of SF-DQN and DQN algorithms by evaluating their generalization error when transferring knowledge from one task to another. Our results indicate that SF-DQN achieves improved generalization compared to DQN, demonstrating the superiority of SF-DQN in transfer RL.

## 1.1 RELATED WORKS

**Successor features in RL.** In the pioneering work, (Dayan, 1993) introduced the concept of SR, demonstrating that the value function can be decomposed into a reward mapping and a state representation that measures the future state occupancy from a given state, with learning feasibility proof in tabular settings. Subsequently, (Barreto et al., 2017) extended SR from three perspectives: (1) the feature domain of SR is extended from states to state-action pairs, known as SF; (2) DNNs are deployed as function approximators to represent the SF and reward mappings; (3) GPI algorithm is introduced to accelerate policy transfer for multi-tasks. (Barreto et al., 2017; 2018) provided transfer guarantees for Q-learning with SF and GPI in the tabular setting. Furthermore, (Kulkarni et al., 2016b; Zhang et al., 2017) apply SF learning with DNN-based schemes to subgoal identification (Kulkarni et al., 2016b) and robot navigation (Zhang et al., 2017). A comprehensive RL transfer comparison using SF under different assumptions can be found in (Zhu et al., 2023).

**RL with neural networks.** Recent advancements in RL with neural network approximation mainly include the Bellman Eluder dimension (Jiang et al., 2017; Russo & Van Roy, 2013), Neural Tangent

Kernel (NTK) (Yang et al., 2020; Cai et al., 2019; Xu & Gu, 2020; Du et al., 2020), and Besov regularity (Suzuki, 2019; Ji et al., 2022; Nguyen-Tang et al., 2022). However, each of these frameworks has its own limitations. The Eluder dimension exhibits exponential growth even for shallow neural networks (Dong et al., 2021), making it challenging to characterize sample complexity in real-world applications of DRL. The NTK framework linearizes DNNs to bypass the non-convexity derived from the non-linear activation function in neural networks. Nevertheless, it requires using computationally inefficient, extremely wide neural networks (Yang et al., 2020). Moreover, the NTK approach falls short in explaining the advantages of utilizing non-linear neural networks over linear function approximation (Liu et al., 2022; Fan et al., 2020). The Besov space framework (Ji et al., 2022; Nguyen-Tang et al., 2022; Liu et al., 2022; Fan et al., 2020) requires sparsity on neural networks and makes the impractical assumption that the algorithm can effectively identify the global optimum, which is unfeasible for non-convex objective functions involving neural networks.

**Theory of generalization in deep learning.** The theory of generalization in deep learning has been extensively developed in supervised learning, where labeled data is available throughout training. Generalization in learned models necessitates low training error and small generalization gap. However, in DNNs, training errors and generalization gaps are analyzed separately due to their non-convex nature. To ensure bounded generalization, it is common to focus on *one-hidden-layer* neural networks (Safran & Shamir, 2018) in convergence analysis. Existing theoretical analysis tools in supervised learning with generalization guarantees draw heavily from various frameworks, including the Neural Tangent Kernel (NTK) framework (Jacot et al., 2018; Du et al., 2018; Lee et al., 2018), model recovery techniques (Zhong et al., 2017; Ge et al., 2018; Bakshi et al., 2019; Soltanolkotabi et al., 2018; Zhang et al., 2020), and the analysis of structured data (Li & Liang, 2018; Shi et al., 2022; Brutzkus & Globerson, 2021; Allen-Zhu & Li, 2022; Karp et al., 2021; Wen & Li, 2021).

## 2 PRELIMINARIES

In this paper, we address the learning problem involving multiple tasks $\{\mathcal{T}_i\}_{i=1}^n$ and aim to find the optimal policy $\pi_i^\star$ for each task $\mathcal{T}_i$. We begin by presenting the preliminaries for a single task and then elaborate on our algorithm for learning with multiple tasks in the following section.

**Markov decision process and Q-learning.** The Markov decision process (MDP) is defined as a tuple $(\mathcal{S}, \mathcal{A}, \mathcal{P}, r, \gamma)$, where $\mathcal{S}$ is the state space and $\mathcal{A}$ is the set of possible actions. The transition operator $\mathcal{P} : \mathcal{S} \times \mathcal{A} \to \Delta(\mathcal{S})$ gives the probability of transitioning from the current state $s$ and action $a$ to the next state $s'$. The function $r : \mathcal{S} \times \mathcal{A} \times \mathcal{S} \to [-R_{\max}, R_{\max}]$ measures the reward for a given state-action pair. The discount factor $\gamma \in [0, 1)$ determines the significance of future rewards.

For the $i$-th task, the goal of the agent is to find the optimal policy $\pi_i^\star$ with $a_t = \pi_i^\star(s_t)$ at each time step $t$. The aim is to maximize the expected discounted sum of reward as $\sum_{t=0}^\infty \gamma^t \cdot r_i(s_t, a_t, s_{t+1})$, where $r_i$ denotes the reward function for the $i$-th task. For any state-action pair $(s, a)$, we define the action-value function $Q_i^\pi$ given a policy $\pi$ as

$$Q_i^\pi(s, a) = \mathbb{E}_{\pi, \mathcal{P}}\left[ \sum_{t=0}^\infty \gamma^t r_i(s_t, a_t, s_{t+1}) \mid s_0 = s, a_0 = a \right]. \tag{1}$$

Then, the optimal $Q$-function, denoted as $Q_i^{\pi^\star}$ or $Q_i^\star$, satisfies

$$Q_i^\star(s, a) := \max_\pi Q_i^\pi(s, a) = \mathbb{E}_{s'|s,a}\, r_i(s, a, s') + \gamma \max_{a'} Q_i^{\pi^\star}(s', a'), \tag{2}$$

where (2) is also known as the Bellman equation. Through the optimal action-value function $Q_i^\star$, the agent can derive the optimal policy (Watkins & Dayan, 1992; Sutton & Barto, 2018) following

$$\pi_i^\star(s) = \arg\max_a Q_i^\star(s, a). \tag{3}$$

**Deep Q-networks (DQNs).** The DQN utilizes a DNN parameterized with weights $\omega$, denoted as $Q_i(s, a; \omega) : \mathbb{R}^d \to \mathbb{R}$ for the $i$-th task, to approximate the optimal Q-value function $Q_i^\star$ in (2). Specifically, given input feature $x := x(s, a)$, the output of the $L$-hidden-layer DNN is defined as

$$Q_i(s, a; \omega) := \omega_{L+1}^\top / K \cdot \sigma\left( \omega_L^\top \cdots \sigma(\omega_1^\top x) \right), \tag{4}$$

where $x = x(s, a)$ and $\sigma(\cdot)$ is the ReLU activation function, i.e., $\sigma(z) = \max\{0, z\}$.

**Successor feature.** For $i$-the task, suppose the expected one-step reward associated with the transition $(s, a, s')$ can be computed as

$$r_i(s, a, s') = \phi(s, a, s')^\top w_i^\star, \qquad with \quad \phi, w_i^\star \in \mathbb{R}^d, \tag{5}$$

where $\phi$ remains the same for all the task. With the reward function in (5), the Q-value function in (1) can be rewritten as

$$Q_i^\pi(\boldsymbol{s},a) = \mathbb{E}_{\pi,\mathcal{P}}\big[\textstyle\sum_{t=0}^\infty \gamma^t \boldsymbol{\phi}(\boldsymbol{s}_t,a_t,\boldsymbol{s}_{t+1}) \mid (\boldsymbol{s}_0,a_0)\big]^\top \boldsymbol{w}_i^\star := \psi_i^\pi(\boldsymbol{s},a)^\top \boldsymbol{w}_i^\star. \tag{6}$$

Then, the optimal Q function satisfies

$$Q_i^\star(\boldsymbol{s},a) = \mathbb{E}_{\pi_i^\star,\mathcal{P}}\big[\textstyle\sum_{i=0}^\infty \gamma^i \boldsymbol{\phi}(\boldsymbol{s}_i,a_i,\boldsymbol{s}_{i+1}) \mid (\boldsymbol{s}_0,a_0)\big]^\top \boldsymbol{w}_i^\star := \psi_i^\star(\boldsymbol{s},a)^\top \boldsymbol{w}_i^\star. \tag{7}$$

## 3 PROBLEM FORMULATION AND ALGORITHM

**Problem formulation.** Without loss of generality, the data is assumed to be collected from the tasks in the order of $\mathcal{T}_1$ to $\mathcal{T}_n$ during the learning process. The goal is to utilize the collected data for each task, e.g., $\mathcal{T}_j$, and the learned knowledge from previous tasks $\{\mathcal{T}_i\}_{i=1}^{j-1}$ to derive the optimal policy $\pi_j^\star$ for $\mathcal{T}_j$. These tasks share the same environment dynamic but the reward function changes across the task as shown in (5). For each task $\mathcal{T}_i$, we denote its reward as

$$r_i = \boldsymbol{\phi} \cdot \boldsymbol{w}_i^\star, \quad with \quad \|\boldsymbol{\phi}\|_2 \le \phi_{\max}, \tag{8}$$

where $\phi$ is the transition feature across all the tasks and $\boldsymbol{w}_i^\star$ is the reward mapping.

From (7), the learning of optimal Q-function for the $i$-th task is decomposed as two sub-tasks: learning SF $\psi_i^\star(\boldsymbol{s},a)$ and learning reward $\boldsymbol{w}_i^\star$.

**Reward mapping.** To find the optimal $\boldsymbol{w}_i^\star$, we utilize the information from $\boldsymbol{\phi}(\boldsymbol{s},a,\boldsymbol{s}')$ and $r_i(\boldsymbol{s},a,\boldsymbol{s}')$. The value of $\boldsymbol{w}^\star$ can be obtained by solving the optimization problem

$$\min_{\boldsymbol{w}_i} \|r_i - \boldsymbol{\phi} \cdot \boldsymbol{w}_i\|_2. \tag{9}$$

**Successor features.** We use $\psi_i^\pi$ to denote the successor feature for the $i$-th task, and $\psi_i^\pi$ satisfies

$$\psi_i^\pi(\boldsymbol{s},a) = \mathbb{E}_{\boldsymbol{s}'|\boldsymbol{s},a}\, \boldsymbol{\phi}(\boldsymbol{s},a,\boldsymbol{s}') + \gamma \cdot \psi_i^\pi\big(\boldsymbol{s}',\pi(\boldsymbol{s}')\big). \tag{10}$$

The expression given by (10) aligns perfectly with the Bellman equation in (2), where $\phi$ acts as the reward. Therefore, following DQNs, we utilize a function $\psi(\boldsymbol{s},a)$ parameterized using the DNN as

$$\psi_i(\Theta_i;\boldsymbol{s},a) = H\big(\Theta_i;\boldsymbol{x}(\boldsymbol{s},a)\big), \tag{11}$$

where $\boldsymbol{x}: \mathcal{S} \times \mathcal{A} \longrightarrow \mathbb{R}^d$ is the feature mapping of the state-action pair. Without loss of generality, we assume $|\boldsymbol{x}(\boldsymbol{s},a)| \le 1$. Then, find $\psi^\star$ is to minimize the mean squared Bellman error (MSBE)

$$\min_{\Theta_i} : f(\Theta_i) := \mathbb{E}_{(\boldsymbol{s},a)\sim\pi^\star}\big[\mathbb{E}_{\boldsymbol{s}'|\boldsymbol{s},a}\,\psi_i(\Theta_i;\boldsymbol{s},a) - \boldsymbol{\phi}(\boldsymbol{s},a,\boldsymbol{s}') - \gamma \cdot \psi_i\big(\Theta_i;\boldsymbol{s}',\pi^\star(s')\big)\big]^2. \tag{12}$$

It is worth mentioning that although (12) and (9) appear to be independent of each other, the update of $\boldsymbol{w}_i$ does affect the update of $\psi_i$ through the shift in data distribution. The collected data is estimated based on the policy depending on the current estimated values of $\psi_i$ and $\boldsymbol{w}_i$, which shifts the distribution of the collected data away from $\pi_i^\star$. This, in turn, leads to a bias depending on the value of $\boldsymbol{w}_i$ in the calculation of the gradient of $\Theta_i$ in minimizing (12).

**Generalized policy improvement (GPI).** Suppose we have acquired knowledge about the optimal successor features for the previous $n$ tasks, and we use $\hat{\psi}_i$ to denote the estimated successor feature function for the $i$-th task. Now, let's consider a new task $\mathcal{T}_{n+1}$ with the reward function defined as $r_{n+1} = \boldsymbol{\phi}\boldsymbol{w}_{n+1}^\star$. Instead of training from scratch, we can leverage the knowledge acquired from previous tasks to improve our approach. We achieve this by deriving the policy follows

$$\pi(a|\boldsymbol{s}) = \arg\max_a \max_{1\le i\le n+1} \hat{\psi}_i(\boldsymbol{s},a)^\top \boldsymbol{w}_{n+1}^\star. \tag{13}$$

This strategy tends to yield better performance than relying solely on $\hat{\psi}_{n+1}(\boldsymbol{s},a)^\top \boldsymbol{w}_{n+1}^\star$, especially when $\hat{\psi}_{n+1}$ has not yet converged to the optimal successor feature $\psi_{n+1}^\star$ during the early learning stage, while some task is closely related to the new tasks, i.e., some $\boldsymbol{w}_i^\star$ is close to $\boldsymbol{w}_{n+1}^\star$. This policy improvement operator is derived from Bellman's policy improvement theorem (Bertsekas & Tsitsiklis, 1996) and (2). When the reward is fixed across different policies, e.g., $\{\pi_i\}_{i=1}^n$, and given that the optimal Q-function represents the maximum across the entire policy space, the maximum of multiple Q-functions corresponding to different policies, $\max_{1\le i\le n} Q^{\pi_n}$, is expected to be closer to $Q^\star$ than any individual Q-function, $Q^{\pi_i}$. In this paper, the parameter $\phi$ in learning the successor feature is analogous to the reward in learning the Q-function. As $\phi$ remains the same for different tasks, this analogy has inspired the utilization of GPI in our setting, even where the rewards change.

### 3.1 SUCCESSOR FEATURE DEEP Q-NETWORK

The goal is to find $\boldsymbol{w}_i$ and $\Theta_i$ by solving the optimization problems in (9) and (12) for each task sequentially, and the optimization problems are solved by mini-batch stochastic gradient descent (mini-batch SGD). Algorithm 1 contains two loops, and the outer loop number $n$ is the number of tasks and inner loop number $T$ is the maximum number of iterations in solving (9) and (12) for each task. At the beginning, we initialize the parameters as $\Theta^{(0)}$ and $\boldsymbol{w}_i^{(0)}$ for task $i$ with $1 \le i \le n$. In $t$-th inner loop for the $i$-th task, let $\boldsymbol{s}_t$ be the current state, and $\theta_c$ be the learned weights for task $c$. The agent selects and executes actions according to

$$a = \pi_\beta(\max_{c \in [i]} \psi(\Theta_c; \boldsymbol{s}_t, a)^\top \boldsymbol{w}_i^{(t)}), \tag{14}$$

where $\pi_\beta(Q(\boldsymbol{s}_t, a))$ is the policy operator based on the function $Q(\boldsymbol{s}_t, a)$, e.g., greedy, $\varepsilon$-greedy, and softmax. For example, if $\pi_\beta(\cdot)$ stands for greedy policy, then $a = \arg\max_a \max_{c \in [i]} \psi(\Theta_c; \boldsymbol{s}_t, a)^\top \boldsymbol{w}_i^{(t)}$. The collected data are stored in a replay buffer with size $N$. Then, we sample a mini-batch of samples from the replay buffer and denote the samples as $\mathcal{D}_t$.

---

**Algorithm 1** Successor Feature Deep Q-Network (SF-DQN)

---

1: **Input**: Number of iterations $T$, and experience replay buffer size $N$, step size $\{\eta_t, \kappa_t\}_{t=1}^T$.
2: Initialize $\{\Theta_i^{(0)}\}_{i=1}^n$ and $\{\boldsymbol{w}_i^{(0)}\}_{i=1}^n$.
3: **for** Task $i = 1, 2, \cdots, n$ **do**
4:      **for** $t = 0, 1, 2, \cdots, T-1$ **do**
5:          Collect data and store in the experience replay buffer $\mathcal{D}_t$ following a behavior policy $\pi_t$ in (14).
6:          Perform gradient descent steps on $\Theta_i^{(t)}$ and $\boldsymbol{w}^{(t)}$ following (15).
7:      **end for**
8:      Return $Q_i = \psi_i(\Theta_i^{(T)})^\top \boldsymbol{w}_i^{(T)}$ for $i = 1, 2, \cdot, n$.
9: **end for**

---

Next, we update the current weights using a mini-batch gradient descent algorithm following

$$\boldsymbol{w}^{(t+1)} = \boldsymbol{w}^{(t)} - \kappa_t \cdot \sum_{m \in \mathcal{D}_t} \left( \boldsymbol{\phi}(\boldsymbol{s}_m, a_m, \boldsymbol{s}_m')^\top \boldsymbol{w}^{(t)} - r(\boldsymbol{s}_m, a_m, \boldsymbol{s}_m') \right) \cdot \boldsymbol{\phi}(\boldsymbol{s}_m, a_m, \boldsymbol{s}_m')$$

$$\Theta_i^{(t+1)} = \Theta_i^{(t)} - \eta_t \cdot \sum_{m \in \mathcal{D}_t} \left( \psi(\Theta_i^{(t)}; \boldsymbol{s}_m, a_m) - \boldsymbol{\phi}(\boldsymbol{s}_m, a_m, \boldsymbol{s}_m') - \gamma \cdot \psi(\Theta_i^{(t)}; \boldsymbol{s}_m', a') \right) \tag{15}$$

$$\cdot \nabla_{\Theta_i} \psi(\Theta_i^{(t)}; \boldsymbol{s}_m, a_m),$$

where $\eta_t$ and $\kappa_t$ are the step sizes, and $a' = \arg\max_a \max_{c \in [i]} \psi(\Theta_c; \boldsymbol{s}_m', a)^\top \boldsymbol{w}_i^{(t)}$. The gradient for $\Theta_i^{(t)}$ in (15) can be viewed as the gradient of

$$\sum_{(\boldsymbol{s}_m, a_m) \sim \mathcal{D}_t} \left( \psi_i(\Theta_i; \boldsymbol{s}, a) - \boldsymbol{\phi} - \mathbb{E}_{\boldsymbol{s}'|\boldsymbol{s}, a} \max_{a'} \psi_i(\Theta_i^{(t)}; \boldsymbol{s}', a') \right)^2, \tag{16}$$

which is the approximation to (12) via replacing $\max_{a'} \psi_i^\star$ with $\max_{a'} \psi_i(\Theta_i^{(t)})$.

## 4 THEORETICAL RESULTS

### 4.1 SUMMARY OF MAJOR THEORETICAL FINDINGS

To the best of our knowledge, our results in Section 4.3 provide the first theoretical characterization for SF-DQN with GPI, including a comparison with the conventional Q-learning under commonly used assumptions. Before formally presenting them, we summarize the highlights as follows.

Table 1: Important Notations

| | | | |
|---|---|---|---|
| $K$ | Number of neurons in the hidden layer. | $L$ | Number of the hidden layers. |
| $d$ | Dimension of the feature mapping of $(\boldsymbol{s}, a)$. | $T$ | Number of iterations. |
| $\Theta_i^\star, \boldsymbol{w}_i^\star$ | The global optimal to (12) and (9) for $i$-th task. | $N$ | Replay buffer size. |
| $\rho_1$ | The smallest eigenvalue of $\mathbb{E}\nabla\boldsymbol{\psi}_i(\Theta_i^\star)\nabla\boldsymbol{\psi}_i(\Theta_i^\star)^\top$. | $\rho_2$ | The smallest eigenvalue of $\mathbb{E}\boldsymbol{\phi}(\boldsymbol{s}, a)\boldsymbol{\phi}(\boldsymbol{s}, a)^\top$. |
| $q^\star$ | A variable indicates the relevance between current and previous tasks. | $C^\star$ | A constant related to the distribution shift between the behavior and optimal policies. |

**(T1) Leaned Q-function converges to the optimal Q-function at a rate of $1/T$ with generalization guarantees.** We demonstrate that the learned parameters $\Theta_i^{(T)}$ and $\boldsymbol{w}_i^{(T)}$ converge towards their respective ground truths, $\Theta_i^\star$ and $\boldsymbol{w}_i^\star$, indicating that SF-DQN converges to optimal Q-function at a rate of $1/T$ as depicted in (23) (Theorem 1). Moreover, the generalization error of the learned

Q-function scales on the order of $\frac{\|\boldsymbol{w}^{(0)} - \boldsymbol{w}^\star\|_2}{1 - \gamma - \Omega(N^{-1/2}) - \Omega(C^\star)} \cdot \frac{1}{T}$. By employing a large replay buffer $N$, minimizing the data distribution shift factor $C^\star$, and improving the estimation of task-specific reward weights $\boldsymbol{w}^{(0)}$, we can achieve a lower generalization error.

**(T2) GPI enhances the generalization of the learned model with respect to the task relevance factor $q^\star$.** We demonstrate that, when GPI is employed, the learned parameters exhibit improved estimation error with a reduction rate at $\frac{1-c}{1-c \cdot q^\star}$ for some constant $c < 1$ (Theorem 2), where $q^\star$ is defined in (24). From (24), it is clear that $q^\star$ decreases as the distances between task-specific reward weights, denoted as $\|\boldsymbol{w}_j^\star - \boldsymbol{w}_i^\star\|_2$, become smaller. This indicates a close relationship between the previous tasks and the current task, resulting in a smaller $q^\star$ and, consequently, a larger improvement through the usage of GPI.

**(T3) SF-DQN achieves a superior performance over conventional DQN by a factor of $\frac{1+\gamma}{2}$ for the estimation error of the optimal Q-function.** When we directly transfer the learned knowledge of the Q-function to a new task without any additional training, our results demonstrate that SF-DQN always outperforms its conventional counterpart, DQN, by a factor of $\frac{1+\gamma}{2}$ (Theorems 3 and 4). As $\gamma$ approaches one, we raise the emphasis on long-term rewards, making the accumulated error derived from the incorrect Q-function more signficant. Consequently, this leads to reduced transferability between the source tasks and the target task. Conversely, when $\gamma$ is small, indicating substantial potential for transfer learning between the source and target tasks, we observe a more significant improvement when using SF-DQN.

## 4.2 ASSUMPTIONS

We propose the assumptions in deriving our major theoretical results. These assumptions are commonly used in existing RL and neural network learning theories to simplify the presentation.

**Assumption 1.** *There exists a deep neural network with weights $\Theta_i^\star$ such that it minimizes (12) for the $i$-th task, i.e, $f(\Theta_i^\star) = 0$.*

Assumption 1 assumes a substantial expressive power of the deep neural network, allowing it to effectively represent $\psi^\star$ in the presence of an unknown ground truth $\Theta^\star$.

**Assumption 2.** *At any fixed outer iteration $t$, the behavior policy $\pi_t$ and its corresponding transition kernel $\mathcal{P}_t$ satisfy*

$$\sup_{\boldsymbol{s} \in \mathcal{S}} \ d_{TV}\big(\mathbb{P}(\boldsymbol{s}_\tau \in \cdot) \mid \boldsymbol{s}_0 = \boldsymbol{s}), \mathcal{P}_t\big) \leq \lambda \nu^\tau, \quad \forall \tau \geq 0 \tag{17}$$

*for some constant $\lambda > 0$ and $\nu \in (0, 1)$, where $d_{TV}$ denotes the total-variation distance.*

Assumption 2 assumes the Markov chain $\{\boldsymbol{s}_n, a_n, \boldsymbol{s}_{n+1}\}$ induced by the behavior policy is uniformly ergodic with the corresponding invariant measure $\mathcal{P}_t$. This assumption is standard in Q-learning (Xu & Gu, 2020; Zou et al., 2019; Bhandari et al., 2018), where the data are non-i.i.d.

**Assumption 3.** *For any $\Theta^{(t,0)} \in \mathbb{R}^n$ and $\boldsymbol{w}^{(t,0)} \in \mathbb{R}^d$, the greedy policy $\pi_t$ at the $t$-th outer loop, i.e., $\pi_t(a|\boldsymbol{s}) = \arg\max_{a'} Q_t(\boldsymbol{s}, a')$, satisfies*

$$\left| \pi_t(a|\boldsymbol{s}) - \pi^\star(a|\boldsymbol{s}) \right| \leq C \cdot \sup_{(\boldsymbol{s},a)} \|Q_t(\boldsymbol{s}, a) - Q^\star(\boldsymbol{s}, a)\|_F, \tag{18}$$

*where $C$ is a positive constant. Equivalently, when $Q_t = \psi(\Theta^{(t)})^\top \boldsymbol{w}^{(t)}$, we have*

$$\left| \pi_t(a|\boldsymbol{s}) - \pi^\star(a|\boldsymbol{s}) \right| \leq C \cdot \big(\|\Theta^{(t)} - \Theta^\star\|_2 + \|\boldsymbol{w}^{(t)} - \boldsymbol{w}^\star\|_2\big). \tag{19}$$

Assumption 3 indicates the policy difference between the behavior policy and the optimal policy. Moreover, (19) can be considered as a more relaxed variant of condition (2) in Zou et al. (2019) as (19) only necessitates the constant to hold for the distance of an arbitrary function from the ground truth, rather than the distance between two arbitrary functions.

## 4.3 MAIN THEORETICAL FINDINGS

### 4.3.1 CONVERGENCE ANALYSIS OF SF-DQN

Theorem 1 demonstrates that the learned Q function converges to the optimal Q function when using SF-DQN for Task 1. Notably, GPI is not employed for the initial task, as we lack prior knowledge

about the environment. Specifically, given conditions (i) the initial weights for $\psi$ are close to the ground truth as shown in (20), (ii) the replay buffer is large enough as in (21), and (iii) the distribution shift between the behavior policy and optimal policy is bounded (as shown in Remark), the learned parameters from Algorithm (1) for task 1, $\psi_1(\Theta_1)$ and $\boldsymbol{w}_1$, converge to the ground truth $\psi_1^\star$ and $\boldsymbol{w}_1^\star$ as in (22), indicating that the learned Q function converges to the optimal Q function as in (23).

**Theorem 1** (Convergence analysis of SF-DQN without GPI). *Suppose the assumptions in Section 4.2 hold and the initial neuron weights of the SF of task 1 satisfy*

$$\frac{\|\Theta_1^{(0)} - \Theta_1^\star\|_F}{\|\Theta_1^\star\|_F} \leq (1 - c_N) \cdot \frac{\rho_1}{K^2}, \tag{20}$$

*for some positive $c_N$. When we select the step size as $\eta_t = \frac{1}{t+1}$, and the size of the replay buffer is*

$$N = \Omega(c_N^{-2}\rho_1^{-1} \cdot K^2 \cdot L^2 d \log q). \tag{21}$$

*Then, with the high probability of at least $1 - q^{-d}$, the weights $\theta^{(T)}$ from Algorithm 1 satisfy*

$$\|\Theta_1^{(T)} - \Theta_1^\star\|_2 \leq \frac{C_1 + C^\star \cdot \|\boldsymbol{w}_1^{(0)} - \boldsymbol{w}_1^\star\|_2}{(1-\gamma-c_N)(1-\gamma)\rho_1 - C^\star} \cdot \frac{\log^2 T}{T},$$

$$\|\boldsymbol{w}_1^{(T)} - \boldsymbol{w}_1^\star\|_2 \leq \left(1 - \frac{\rho_2}{\phi_{\max}}\right)^T \|\boldsymbol{w}_1^{(0)} - \boldsymbol{w}_1^\star\|_2, \tag{22}$$

*where $C_1 = (2 + \gamma) \cdot R_{\max}$, and $C^\star = |\mathcal{A}| \cdot R_{\max} \cdot (1 + \log_\nu \lambda^{-1} + \frac{1}{1-\nu}) \cdot C$. Specifically, the learned Q-function satisfies*

$$\max_{\boldsymbol{s},a} \left|Q_1 - Q^\star\right| \leq \frac{C_1 + \|\boldsymbol{w}_1^{(0)} - \boldsymbol{w}_1^\star\|_2}{(1-\gamma-c_N)(1-\gamma)\rho_1 - 1} \cdot \frac{\log^2 T}{T} + \frac{\|\boldsymbol{w}_1^{(0)} - \boldsymbol{w}_1^\star\|_2 R_{\max}}{1-\gamma}\left(1 - \frac{\rho_2}{\phi_{\max}}\right)^T. \tag{23}$$

**Remark 1** (upper bound of $C$): To ensure the meaningfulness of the upper bound in (23), specifically that the denominator needs to be greater than 0, $C$ has an explicit upper bound as $C \leq \frac{(1-\gamma-c_N)(1-\gamma)\rho_1}{|\mathcal{A}| \cdot R_{\max}}$. Considering the definition of $C$ in Assumption 3, it implies that the difference between the behavior policy and the optimal policy is bounded. In other words, the fraction of bad tuples in the collected samples is constrained.

**Remark 2** (Initialization): Note that (20) requires a good initialization. Firstly, it is still a state-of-the-art practice in analyzing Q-learning via deep neural network approximation. Secondly, according to the NTK theory (Jacot et al., 2018), there always exist some good local minima, which is almost as good as the global minima, near some random initialization. Finally, such a good initialization can also be adapted from some pre-trained models.

### 4.3.2 IMPROVED PERFORMANCE WITH GENERALIZED POLICY IMPROVEMENT

Theorem 2 establishes that the estimated Q function converges towards the optimal solution with the implementation of GPI as shown in (25), leveraging the prior knowledge learned from previous tasks. The enhanced performance associated with GPI finds its expression as $q^\star$ defined in (24). Notably, when tasks $i$ and $j$ exhibit a higher degree of correlation, meaning that the distance between $\boldsymbol{w}_i^\star$ and $\boldsymbol{w}_j^\star$ for tasks $i$ and $j$ is minimal, we can observe a more substantial enhancement by employing GPI in the process of transferring knowledge from task $i$ to task $j$ from (25).

**Theorem 2** (Convergence analysis of SF-DQN with GPI). *Let us define*

$$q^\star = \frac{(1+\gamma)R_{\max}}{1-\gamma} \cdot \frac{\min_{1 \leq i \leq j-1} \|\boldsymbol{w}_i^\star - \boldsymbol{w}_j^\star\|_2}{\|\Theta_j^{(0)} - \Theta_j^\star\|_2}. \tag{24}$$

*Then, with the probability of at least $1 - q^{-d}$, the neuron weights $\Theta_j^{(T)}$ for the $j$-th task satisfy*

$$\|\Theta_j^{(T)} - \Theta_j^\star\|_2 \leq \frac{C_1 + C^\star \|\boldsymbol{w}_j^{(0)} - \boldsymbol{w}_j^\star\|_2}{(1-\gamma-c_N)(1-\gamma)\rho_1 - \min\{q^\star, 1\} \cdot C^\star} \cdot \frac{\log^2 T}{T}. \tag{25}$$

**Remark 3 (Improvement via GPI)**: Utilizing GPI enhances the convergence rate from in the order of $\frac{1}{1-C^\star} \cdot \frac{1}{T}$ to in the order of $\frac{1}{1-q^\star \cdot C^\star} \cdot \frac{1}{T}$. When the distance between the source task and target tasks is small, $q^\star$ can approach zero, indicating an improved generalization error by a factor of $1 - C^\star$, where $C^\star$ is proportional to the fraction of bad tuples. The improvement achieved through GPI is derived from the reduction of the distance between the behavior policy and the optimal policy, subsequently decreasing the fraction of bad tuples in the collected data. Here, $C^\star$ is proportional to the fraction of bad tuples without using GPI, and $q^\star \cdot C^\star$ is proportional to the fraction of bad tuples when GPI is employed.

### 4.3.3 BOUNDS FOR TRANSFER REINFORCEMENT LEARNING

From Theorems 1 and 2, we have successfully estimated $Q_i^{\pi_i^\star}$ for task $i$ using our proposed SF-DQN. When the reward changes to $r_{n+1}(\boldsymbol{s}, a, \boldsymbol{s}') = \boldsymbol{\phi}^\top(\boldsymbol{s}, a, \boldsymbol{s}')\boldsymbol{w}_{n+1}^\star$ for a new task $\mathcal{T}_{n+1}$, as long as we have estimated $\boldsymbol{w}_{n+1}^\star$, we can calculate the estimated Q-value function for $\mathcal{T}_{n+1}$ simply by setting

$$Q_{n+1}^{\pi_{n+1}}(\boldsymbol{s}, a) = \max_{1 \leq j \leq n} \psi(\Theta_j^{(T)}; \boldsymbol{s}, a)\boldsymbol{w}_{n+1}^\star. \tag{26}$$

As $\boldsymbol{w}_{n+1}^{(t)}$ experiences linear convergence to its optimal $\boldsymbol{w}^\star$, which is significantly faster than the sublinear convergence of $\Theta_{(n+1)}^{(t)}$, as shown in (22), this derivation of $Q_{n+1}$ in (26) simplifies the computation of $\Theta_{n+1}^\star$ into a much more manageable supervised setting for approximating $w_{n+1}^\star$ with only a modest performance loss as shown in (27). This is demonstrated in the following Theorem 3.

**Theorem 3** (Transfer learning via SF-DQN). *For the (n+1)-th task with $r_{n+1} = \boldsymbol{\phi}^\top w_{n+1}^\star$, suppose the Q-value function is derived based on (26), we have*

$$\max |Q_{n+1}^{\pi_{n+1}} - Q_{n+1}^\star| \leq \frac{1+\gamma}{1-\gamma}\phi_{\max} \min_{j \in [n]} \|\boldsymbol{w}_j^\star - \boldsymbol{w}_{n+1}^\star\|_2 + \frac{\|\boldsymbol{w}_{n+1}^\star\|_2}{(1-\gamma) \cdot T}. \tag{27}$$

**Remark 4 (Connection with existing works):** The second term of the upper bound in (27), $\frac{\|\boldsymbol{w}_{n+1}^\star\|_2}{(1-\gamma)\cdot T}$, can be explained as $\epsilon$ in Barreto et al. (2017), which results from the approximation error of the optimal Q-functions in the previous tasks.

Without the SF decomposition as shown in (7), one can apply a similar strategy in (26) for DQN as

$$Q_{n+1}^{\pi'_{n+1}}(s, a) = \max_{1 \leq j \leq n} Q(\omega_j^{(T)}; s, a). \tag{28}$$

In Theorem 4, (29) illustrates the performance of (28) through DQN. Compared to Theorem 3, transfer learning via DQN is worse than that via SF-DQN by a factor of $\frac{1+\gamma}{2}$ when comparing the estimation error of the optimal function $Q_{n+1}^\star$ in (27) and (29), indicating the advantages of using SFs in transfer reinforcement learning.

**Theorem 4** (Transfer learning via DQN). *For the (n+1)-th task with $r_{n+1} = \boldsymbol{\phi} \cdot w_{n+1}^\star$, suppose the Q-value function is derived based on (28), we have*

$$\max |Q_{n+1}^{\pi'_{n+1}} - Q_{n+1}^\star| \leq \frac{2}{1-\gamma}\phi_{\max} \cdot \min_{j \in [n]} \|\boldsymbol{w}_j^\star - \boldsymbol{w}_{n+1}^\star\|_2 + \frac{\|\boldsymbol{w}_{n+1}^\star\|_2}{(1-\gamma) \cdot T}. \tag{29}$$

**Remark 5 (Improvement by a factor of $\frac{1+\gamma}{2}$):** Transfer learning performance in SF-DQN is influenced by the knowledge gap between previous and current tasks, primarily attributed to differences in rewards and data distribution. In SF-DQN, the impact of reward differences is relatively small since $\phi$ that plays the role of reward remains fixed. The parameter $\gamma$ affects the influence of data distribution differences. A small $\gamma$ prioritizes immediate rewards, thereby the impact of data distribution on the knowledge gap is not significant. With a small $\gamma$, the impact of reward difference dominates, resulting in a high gap between SF-DQN and DQN in transfer learning.

### 4.4 TECHNICAL CHALLENGES, COMPARISON WITH EXISTING WORKS

**Beyond deep learning theory: Challenges in deep reinforcement learning.** The proof of Theorem 1 is inspired from the convergence analysis of one-hidden-layer neural networks within the (semi-)supervised learning domain (Zhong et al., 2017; Zhang et al., 2022). This proof tackles *two primary objectives*: i) the first objective involves characterizing the local convex region of the objective functions presented in (12) and (9); ii) the second objective focuses on quantifying the distance between the gradient defined in (15) and the gradient of the objective functions in (12) and (9).

However, extending this approach from the (semi-)supervised learning setting to the deep reinforcement learning domain introduces *additional challenges*. First, we expand our proof beyond the scope of one-hidden-layer neural networks to encompass multi-layer neural networks. This extension requires new technical tools for characterizing the Hessian matrix and concentration bounds, as outlined in Appendix F.1. Second, the approximation error bound deviates from the supervised learning scenarios due to several factors: the non-i.i.d. of the collected data, the distribution shift between the behavior policy and the optimal policy, and the approximation error incurred when utilizing (16) to estimate (12). Addressing these challenges requires developing supplementary tools, as mentioned in Lemma 7. Notably, this approximation does not exhibit scaling behavior proportional to $\|\Theta_i - \Theta_i^\star\|_2$, resulting in a sublinear convergence rate.

**Beyond DQN: challenges in GPI.** The major challenges in proving Theorems 2-4 centers on deriving the improved performance by utilizing GPI. The intuition is as follows. Imagine we have two

closely related tasks, labeled as $i$ and $j$, with their respective optimal weight vectors, $\boldsymbol{w}_i^\star$ and $\boldsymbol{w}_j^\star$, being close to each other. This closeness suggests that these tasks share similar rewards, leading to a bounded distributional shift in the data, which, in turn, implies that their optimal Q-functions should exhibit similarity. To rigorously establish this intuition, we aim to characterize the distance between these optimal Q-functions, denoted as $|Q_i^\star - Q_j^\star|$, in terms of the Euclidean distance between their optimal weight vectors, $\|\boldsymbol{w}_i^\star - \boldsymbol{w}_j^\star\|_2$ (See details in Appendix G). Furthermore, we can only estimate the optimal Q-function for previous tasks during the learning process, and such an estimation error accumulates in the temporal difference learning, e.g., the case of the SF learning of $\psi^\star$. We need to develop novel analytical tools to quantify the error accumulating in the temporal difference learning (see details in Appendix C), which is unnecessary for supervised learning problems.

## 5 EXPERIMENTS

This section summarizes empirical validation for the theoretical results obtained in Section 4 using a synthetic RL benchmark environment. The experiment setup and additional experimental results for real-world RL benchmarks are summarized in Appendix E.

**Convergence of SF-DQN with varied initialization.** Figure 1 shows the performance of Algorithm 1 with different initial $\boldsymbol{w}_1^{(0)}$ to the ground truth $\boldsymbol{w}_1^\star$. When the initialization is close to the ground truth, we observe an increased accumulated reward, which verifies our theoretical findings in (23) that the estimation error of the optimal Q-function reduces as $\|\boldsymbol{w}_1^{(0)} - \boldsymbol{w}^\star\|_2$ decreases.

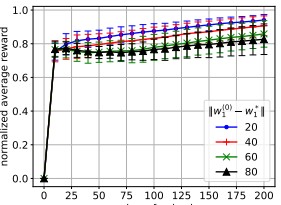
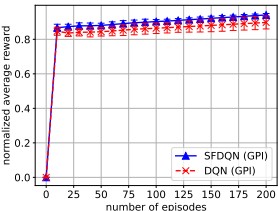

Figure 1: Performance of SF-DQN presented in Algorithm 1 on Task 1.

Figure 2: Transfer comparison for SF-DQN and DQN (with GPI)

**Performance of SF-DQN with GPI when adapting to tasks with varying relevance.** We conducted experiments to investigate the impact of GPI with varied task relevance. Since the difference in reward mapping impacts data distribution shift, rewards, and consequently the optimal Q-function, we utilize the metric $\|\boldsymbol{w}_1^\star - \boldsymbol{w}_2^\star\|_2$ to measure the task irrelevance. The results summarized in Table 2 demonstrate that when tasks are similar (i.e., small $\|\boldsymbol{w}_1^\star - \boldsymbol{w}_2^\star\|$), SF-DQN with GPI consistently outperforms its counterpart without GPI. However, when tasks are dissimilar (i.e., large $\|\boldsymbol{w}_1^\star - \boldsymbol{w}_2^\star\|$), both exhibit same or similar performance, indicating that GPI is ineffective when two tasks are irrelevant. The observations in Table 2 validate our theoretical findings in (25), showing a more significant improvement in using GPI as $\|\boldsymbol{w}_1^\star - \boldsymbol{w}_2^\star\|_2$ decreases.

Table 2: Normalized average reward for SF-DQN with and without GPI.

| $\|w_1^* - w_2^*\|$ | $= 0.01$ | $= 0.1$ | $= 1$ | $= 10$ |
|---|---|---|---|---|
| SF-DQN (w/ GPI) | $\mathbf{0.986} \pm 0.007$ | $\mathbf{0.965} \pm 0.007$ | $\mathbf{0.827} \pm 0.008$ | $\mathbf{0.717} \pm 0.012$ |
| SF-DQN (w/o GPI) | $0.942 \pm 0.004$ | $0.911 \pm 0.013$ | $0.813 \pm 0.009$ | $0.707 \pm 0.011$ |

**Comparison of the SF-DQN agent and DQN agent.** From Figure 2, it is evident that the SF-DQN agent consistently achieves a higher average reward (task 2) than the DQN when starting training on task 2, where transfer learning occurs. These results strongly indicate the improved performance of the SF-DQN agent over the DQN, aligning with our findings in (27) and (29). SF-DQN benefits from reduced estimation error of the optimal Q-function compared to DQN when engaging in transfer reinforcement learning for relevant tasks.

## 6 CONCLUSION

This paper analyzes the transfer learning performance of SF & GPI, with SF being learned using deep neural networks. Theoretically, we present a convergence analysis of our proposed SF-DQN with generalization guarantees and provide theoretical justification for its superiority over DQN without using SF in transfer reinforcement learning. We further verify our theoretical findings through numerical experiments conducted in both synthetic and benchmark RL environments. Future directions include exploring the possibility of learning $\phi$ using a DNN approximation and exploring the combination of successor features with other deep reinforcement learning algorithms.

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
