# OpenReview forum: "Provable Knowledge Transfer using Successor Feature for Deep Reinforcement Learning"
_ICLR.cc/2024/Conference — Submitted to ICLR 2024_

### Official Review · Reviewer_qMSi · 2023-10-31

**Soundness:** 2 fair
**Presentation:** 2 fair
**Contribution:** 3 good
**Rating:** 3
**Confidence:** 3

**Summary:**

The paper investigates theoretical properties of the combination of SF-DQN + GPI algorithm for multi-task reinforcement learning. New theoretical results show that the above combination leads to better performance by reusing prior tasks using GPI.

**Strengths:**

1. The paper proposes new theoretical insights into the Successor feature + GPI algorithm for the deep multitask reinforcement learning setting. The authors prove the convergence of this algorithm at a rate 1/T to the optimal Q function with multi-layer neural network and also quantify the generalization error of the learned Q function.
2. The paper also shows what is the benefit afforded by using the successor feature formulation for transfer learning — the perfomance gap in Q function learned via DQN is worse than a factor of (1+\gamma)/2.
3. Finally authors verify that the trends suggested by their bounds like dependence on initialization of weight vectors and task relevance holds by experiments on a toy domain.

**Weaknesses:**

1. I am doubtful about the significance of convergence results. The convergence result with GPI follows the same rate as the convergence rate without GPI. It is hard to tell directly what is the difference in the constants. Having a thorough discussion with some examples would serve to give readers a better understanding of the upper bound.
2. I have the same question about transfer results, how important is the factor of  (1+\gamma)/2. Why does it make the bound substantially looser? Even then we are comparing upper bounds which are not tight, so what is the correct way to make sense of this result?
3. The experiments seem to present results that seem obvious. If weights are initialized closer to optimal weights convergence is fast and if the weight of transfer task is similar to prior task the transfer is fast. This is also similar to bounds we have seen for successor features in previous works.

**Questions:**

1. On page 5, Equation 15, should the gradient for theta be wrt. the best action across all (tasks) successor features based Q function or just the gradient with best action across the successor feature based Q function of current task.
2. In equation 17, I dont believe the notations are correct or have been explained previously in the paper. What is P(s_tau\in .)? What does v_\tau stand for?
3. How is \phi obtained in all the experiments? I dont think \phi is trained in this work based on the exposition?

---

> ### Author Response · Authors · 2023-11-22
> **Response to Reviewer qMSi**
>
> We thank the reviewer's comprehensive reading of this paper. The point-to-point responses are posted based on the reviewer’s questions.
>
> *(**Q1**). I am doubtful about the significance of convergence results. The convergence result with GPI follows the same rate as the convergence rate without GPI. It is hard to tell directly what is the difference in the constants. Having a thorough discussion with some examples would serve to give readers a better understanding of the upper bound.*
>
> (**A1**.) We appreciate your insightful question. We have included several examples to help understand the improvement of using GPI. In addition, we would like to clarify that the improvement achieved through GPI can be as substantial as ensuring that all the data consists of good tuples, and the improvement with a constant factor is the best one can achieve.
>
> **Firstly, utilizing GPI improves the convergence rate from the order of $\frac{1}{1-C^\star}\cdot \frac{1}{T}$ to the order of $\frac{1}{1-q^\star \cdot C^\star}\cdot \frac{1}{T}$,** where $q^\star$ measures the similarity between the source task and target task. Specifically, $q^\star$ is in the order of $\min_{1\le i \le j-1}|w_i^\star-w_j^\star|$. When the distance between the source task ($w_j$) and target tasks ($w_j$) is small, $q^\star$ can be close to zero, indicating improved estimation (generalization) error by a factor of $1-C^\star$, where $C^\star$ is proportional to the fraction of bad tuples.
>
> **Secondly, the improvement is derived from the utilization of GPI in the theoretical characterization, which reduces the distance between the behavior policy and the optimal policy, thus lowering the fraction of bad tuples in the collected data.** The proof of Theorem 2 presented in Appendix D follows the same steps as the proof of Theorem 1, with the crucial difference that the upper bound for $C_t$ is improved from $1$ to $q_t$ (where $q^\star$ is an upper bound for $q_t$) when using GPI. A reduced upper bound for $C_t$ implies a decreased distribution shift between the behavior policy and the optimal policy (see $I_3$ in Appendix F.2), resulting in a reduced gradient difference between the objective function and the population risk function (equation (50) in Lemma 7), indicating a faster convergence rate.
>
> **Thirdly, we want to emphasize that achieving improvement by a constant factor represents the best possible outcome.** As mentioned in the previous paragraph, the fundamental reason for the improvement using GPI lies in its ability to provide a better behavior policy during data collection, specifically by reducing the occurrence of bad tuples. Here, we can observe that $C^\star$ is proportional to the fraction of bad tuples without using GPI, and $q^\star\cdot C^\star$ is proportional to the fraction of bad tuples when GPI is employed. When $q^\star\cdot C^\star$ equals $0$, representing the best possible case where all tuples are good, one can verify that the convergence rate remains in the order of $\frac{1}{T}$. Thus, using GPI to achieve improvement with a constant factor represents the best attainable outcome.

---

> ### Author Response · Authors · 2023-11-22
> **Response to Reviewer qMSi (Part II)**
>
> *(**Q2**). I have the same question about transfer results, how important is the factor of (1+\gamma)/2. Why does it make the bound substantially looser? Even then we are comparing upper bounds which are not tight, so what is the correct way to make sense of this result?*
>
> (**A2**.) Thank you very much for your insightful questions. To offer a comprehensive understanding of Theorems 3 and 4, we provide the intuition to understand the theorems, an illustration of the theoretical framework used to derive the results in this response.
>
> **Firstly, we would like to clarify the intuition in understanding the bound with the dependence on $\gamma$.**  Intuitively, the performance of transfer learning is influenced by the knowledge gap between previous tasks and the current task. Recall that the goal is to maximize $E_{\pi_i^\star} \sum_{t}\gamma^t r_i(s_t, a_t)$ for a task $\mathcal{T}_i$. The knowledge gap can be attributed to two main factors: (i) the difference in reward functions between the current task and previous tasks, and (ii) the difference in the optimal policy (and thus the corresponding data distribution) between the current task and previous tasks. In the context of SF-DQN, factor (i) has minimal influence because $\phi$ can be considered as the reward when learning the successor feature, and the function $\phi$ remains fixed across all tasks. The parameter $\gamma$ affects the impact of factor (ii). When $\gamma$ is small or close to zero, the emphasis is primarily on immediate rewards, meaning that the influence of data distribution on the knowledge gap between previous and current tasks is minimal. However, as $\gamma$ increases, prioritizing future rewards, the data distribution becomes a significant factor affecting the final knowledge gap. Therefore, factor (ii) does not exhibit a significant difference when using SF-DQN or DQN, as the knowledge gap always exists. When $\gamma$ is small, the influence of factor (i) dominates the knowledge gap, resulting in a significant improvement in the transfer learning performance using SF-DQN compared to DQN. Conversely, when $\gamma$ is large, the knowledge gap derived from factor (ii) dominates, leading to no significant improvement when using SF-DQN over DQN in transfer learning settings.
>
> **Secondly, this paper provides the first theoretical justification that SF-DQN outperforms DQN in transfer reinforcement learning settings.** Notably, since $\gamma$ is always smaller than 1, the factor $(1+\gamma)/2$ is consistently less than 1. As a result, the upper bound of transfer learning using SF-DQN is consistently superior to DQN. As far as we know, there are no comparable results in existing literature for transfer learning using DQN, let alone for SF-DQN, or the direct comparison between SF-DQN and DQN. While our analysis focuses only on the upper bound, no comparable theoretical results are observed in the existing literature.
>
> **Thirdly, the improvement of SF-DQN arises from the decomposition of the Q-function into the successor feature function and the reward mapping function, which are absent in DQN.** Both proofs follow similar steps, except for bounding the difference between the generalized policy and individual policies, as illustrated in (67) for SF-DQN and (75) for DQN. Compared with (75), (67) in SF-DQN lacks the factor of $\frac{1}{1-\gamma}\cdot \phi_{\max}\cdot||w_{n+1}^\star-w_j^\star||_2$ due to our ability to learn the reward mapping and successor feature separately, mitigating the influence of changes in data distribution. Furthermore, all other constants remain consistent, as we follow the exact proof steps for Theorems 3 and 4. The proofs for Theorems 3 and 4 are included in Appendix C.
>
> **Finally, our theoretical results justified the observation of SF-DQN over DQN in transfer reinforcement learning in real applications.** Our manuscript (Appendix E.2 in the supplementary) presents enlightening empirical findings that are obtained based on real-data experiments using the SF-DQN and DQN. We showed that SF-DQN achieves a better performance than DQN when transferring learning from the source tasks to the target task. In addition, similar observations are presented in (Barreto et.al, 2017)  for SF-DQN and DQN.

---

> ### Author Response · Authors · 2023-11-22
> **Response to qMSi (Part III)**
>
> *(**Q3**) The experiments seem to present results that seem obvious. If weights are initialized closer to optimal weights convergence is fast and if the weight of transfer task is similar to prior task the transfer is fast. This is also similar to bounds we have seen for successor features in previous works.*
>
> (**A3**.) we would like to clarify that there are some fundamental differences between our work and (Barreto et al., 2017) although we share some similar observations. Here, we suspect that the previous works are referring to the representative work of successor features (Barreto et al., 2017). If the reviewers suggest comparing our work with other studies, we are open to providing further clarification.
>
> **Firstly, previous work (Barreto et al., 2017) did not provide the theoretical justification or empirical justification for the case that the weights are initialized closer to optimal weights.** (Barreto et al. 2017) assume that mapping weights $w^star_i$ are known, and therefore they did not provide any conclusion related to the initialization of the weights. While in our manuscript, we assume $w^\star_i$ is unknown and we need to learn it. Therefore, there are no similar results that “weights are initialized closer to optimal weights convergence is fast” presented in (Barreto et al., 2017).
>
> **Secondly, this paper provides both convergence analysis and sample complexity for successor feature learning, while previous work by Barreto et al. (2017) did not include such analyses.** The previous work by Barreto et al. (2017) did not include a convergence analysis. Additionally, in their analysis of transfer learning, Barreto et al. (2017) did not provide a comparison with DQN. Furthermore, the convergence analysis for SF-DQN is challenging due to the non-convex nature of the objective function involving deep neural networks, and existing learning theory primarily focuses on shallow neural networks.
>
> **Finally, although these observations seem to be intuitive, our paper provides the first theoretical justification for these numerical observations.** To the best of our knowledge, our manuscript provides (i) the first convergence analysis of SF-DQN to the optimal Q-function with generalization guarantee, (ii) the first theoretical characterization of enhanced performance using GPI, and (iii) the first theoretical characterization of the superior performance with SF-DQN over DQNs.  As elaborated in Section 4.4, we encountered several challenges in deriving these results. These challenges involved extending the deep learning theory from the supervised learning setting to the successor feature Q-learning setting, as well as adapting the GPI theorem to the context of successor feature learning.
>
> *(**Q4**.) On page 5, Equation 15, should the gradient for theta be wrt. the best action across all (tasks) successor features based Q function or just the gradient with best action across the successor feature based Q function of the current task.*
>
> (**A4**.) Thanks for pointing out our typo. The gradient of theta should be w.r.t. The best action across all successor feature-based Q-functions, and we have revised it accordingly in the paper.
>
> $$\Theta_i^{(t+1)}= \Theta_i^{(t)} - \eta_t\cdot\sum_{m\in\mathcal{D}_t} \Big(\psi(\Theta_i^{(t)};s_m,a_m) -\boldsymbol{\phi}(s_m,a_m,s_m^\prime) -\gamma \cdot \psi(\Theta_i^{(t)};s_m',a_m') \Big)$$
>
> where $a_m^\prime = \pi_\beta(\max_{c\in[i]}\psi(\hat{\Theta}_c;s, a)^\top w_i^{(t)}|s_m^\prime)$.
>
>
> *(**Q5**.) In equation 17, I don't believe the notations are correct or have been explained previously in the paper. What is P(s_tau\in .)? What does v_\tau stand for?*
>
>
> **A5**. We apologize for any confusion caused by the notations. Equation 17 is derived from a standard assumption in TD learning with function approximation, and thus, we did not provide further clarification for it. The term  $P(s_\tau \in .)$ represents the probability measure of the state after $\tau$ steps following a Markovian chain. While there is no specific notation for $v_\tau$, we presume that the reviewer is referring to the notation $v^\tau$, which denotes $v$ raised to the power of $\tau$. Please inform us if you need further clarification.
>
> Equation 17 is introduced to control the statistical bias in the gradient updates when the samples are collected from a Markov observation model. Additionally, Equation 17 is a standard assumption in analyzing TD learning with function approximation, see e.g.,  Assumption 1 in (Zou et al., 2019) and Assumption 5.2 in (Xu & Gu et al., 2020) for Q-learning.

---

> ### Author Response · Authors · 2023-11-22
> **Response to Reviewer qMSi (Part IV)**
>
> *(**Q6**.) How is \phi obtained in all the experiments? I don't think \phi is trained in this work based on the exposition.*
>
> (**A6**.) We apologize for omitting the description of obtaining $\phi$.
>
> In the synthetic experiments detailed in Section 5, we derive $\phi$ based on task 1. Subsequently, for the remaining tasks, we maintain the fixed $\phi$ while generating the corresponding reward $r_i$ for the $i$-th task using equation (8) with a (randomly or specifically) generated reward mapping $w_i$.
>
> In the Reacher environment experiments presented in Appendix E, we can theoretically compute the optimal $\phi$, which is the concatenation of the rewards for all the tasks considered in the environment. This aligns with the standard setup outlined in (Barreto et al., 2017).

---

### Official Review · Reviewer_c6vz · 2023-10-31

**Soundness:** 1 poor
**Presentation:** 3 good
**Contribution:** 2 fair
**Rating:** 3
**Confidence:** 3

**Summary:**

This paper studies the multi-task RL evironment where the environments have different reward functions but share the same underlying transition dynamics. The paper proposes a Q-function decomposition and a generalized policy improvement (GPI) algorithm to to find the optimal Q-function. This paper provides theoretical results for the speed of convergence rate and generalization. Numerical experiments are also provided.

**Strengths:**

- transfer learning and successor features are interesting avenues of research
- the claims are ambitious

**Weaknesses:**

- The Theorems do not seem to be correct. For instance, it does not seem possible that the size of a replay memory impacts the bound as it does given that it could all be filled with bad tuples.
- The experiments do not follow all good practice (see questions below)
- There are some typos, e.g. in Assumption 1 "such that minimizes (12) for"

**Questions:**

THEORY
- Q-learning does not have any valid proof of convergence when used in conjunction with deep learning. Can you clarify how divergence is prevented in this case? In the results from the paper, a large enough replay memory ensures good convergence. How can the replay memory size by itself be a good measure of the quality of the tuples that are used? (It could all be filled with bad tuples).

EXPERIMENTS
- Why is there no variance in the results reported for the experiements?
- Why does SFDQN (GPI) has a normalized average reward that decreases with the number of episodes? Given the theory that the paper describes, this should slowly converge towards the optimum.

---

> ### Author Response · Authors · 2023-11-22
> **Response to Reviewer c6vz**
>
> We thank the reviewer's comprehensive reading of this paper. The point-to-point responses are posted based on the reviewer’s questions.
>
> *(**Q1**.) The Theorems do not seem to be correct. For instance, it does not seem possible that the size of a replay memory impacts the bound as it does given that it could all be filled with bad tuples. Can you clarify how divergence is prevented in this case? In the results from the paper, a large enough replay memory ensures good convergence. How can the replay memory size by itself be a good measure of the quality of the tuples that are used? (It could all be filled with bad tuples).*
>
> (**A1**.) Thanks for your insightful questions! The replay memory size is just one condition for ensuring convergence. **As you rightly pointed out, if all tuples are bad, then $1-C^\star$ will become smaller than $1$, leading to divergence.**
>
> **Firstly, $C^\star$ is corresponding to the fraction of bad tuples. $C^\star$ is proportional to the constant $C$ defined in Assumption 3.** $C$ captures that the data distribution generated by the behavior policy and optimal policy should be bounded, and this is proportional to the fraction of unfavorable tuples. Therefore, $C^\star$ is corresponding to the fraction of bad tuples.
>
> **Secondly, $C^\star$ has an implicit upper bound.** Without delving into the mathematical details, this can be inferred from our summarized takeaway. As discussed in our first takeaway (T1) in Section 4.1, we concluded that "the generalization error of the learned Q-function scales on the order of $\frac{||w^{(0)}-w^\star||_2}{1-\gamma - N^{-1/2}-C^\star}$". It is evident that $C^\star$ has an upper bound of $1-\gamma-N^{-1/2}$ (otherwise the error bound is zero ). Without this upper bound, the estimation bound of the Q-function becomes meaningless.
>
> Finally, for more details, we recommend the reviewer check the [General Response (GR)](https://openreview.net/forum?id=s6bKLlF4Pe&noteId=5sCZCZJLVj).
>
>
>
>
> *(**Q2**.) The experiments do not follow all good practice (see questions below). (i) Why is there no variance in the results reported for the experiements? (ii) Why does SFDQN (GPI) has a normalized average reward that decreases with the number of episodes? Given the theory that the paper describes, this should slowly converge towards the optimum.*
>
>
>
> (**A2**.) Thank you very much for pointing out our bad practice in presenting the numerical experiments. Following your suggestions, we have included the error bar (denotes the variance) in the numerical results, including Figures 1&2, and Table 2 in Section 5, and Figure 4 in Appendix E.2 in the supplementary materials. We provide the average and standard deviation over 5 seeds. Note that, due to the tuning of the experiment setup, some numerical values have changed, but the observations are consistent with the prior results. The codes have also been updated in the supplementary materials.
>
> Regarding the normalized average reward decreasing in Figure (2) (for both methods), we found out this is due to our choice of learning rate and epsilon annealing method when implementing respective algorithms. We fine-tuned the learning rates and epsilon annealing for all the methods and rerun all the experiments. Specifically, we tuned down the learning rates from 1 to 0.1 for all methods, which improved the stability in the convergence. Furthermore, we changed the epsilon annealing to start with $\epsilon=0.5$ and rapidly changed it to $\epsilon=0$ by iteration 200, which improved the fast and stable convergence of the methods. Since the choice of epsilon was not the main focus of our analysis, this was not carefully chosen in the previous experiments. However, it was observed this careful choice was needed for the stability of the convergence of all the methods.

---

> ### Author Response · Authors · 2023-11-22
> **Response to Reviewer c6vz (Part II)**
>
> *(**Q3**.) Q-learning does not have any valid proof of convergence when used in conjunction with deep learning.*
>
> (**A3**.) While we acknowledge that proving convergence in the context of deep learning requires additional assumptions due to non-convexity, it is feasible to establish a valid proof of convergence by introducing standard assumptions, which are also essential for convergence analysis in supervised learning settings. For example, [Cai et.al, 2019] and [Xu & Gu, 2020] are pioneering works in showing the global convergence of Temporal-difference (TD) learning with deep neural network approximation. The key strategy involves employing a linear model to approximate the deep neural network, enabling the utilization of techniques based on linear function approximation. To ensure the validity of the approximation error between the neural network and the linear model, specific assumptions are required. These include the existence of a good minimizer near the initialization, restrictions on the movement of weights away from the initialization, and the necessity of a small distribution shift. Importantly, these assumptions are also indispensable for supervised learning problems involving deep neural networks. Therefore, we find the statement that 'Q-learning does not have any valid proof of convergence when used in conjunction with deep learning' to be overly assertive.
>
> [Cai et.al, 2019] Qi Cai, Zhuoran Yang, Jason Lee, and Zhaoran Wang. Neural temporal-difference learning converges to global optima. Advances in Neural Information Processing Systems, 32, 2019.
>
> [Xu & Gu, 2020] Pan Xu and Quanquan Gu. A finite-time analysis of q-learning with neural network function approximation. In International Conference on Machine Learning, pp.10555–10565.PMLR,2020.
>
>
> *(**Q4**.) There are some typos, e.g. in Assumption 1 "such that minimizes (12) for".*
>
> (**A4**.) Thanks for pointing out our typo! The corresponding sentence has been revised as “There exists a deep neural network with weights $\Theta_i^\star$ such that it minimizes (12) for the $i$-th task, i.e, $f(\Theta_i^\star) = 0$.” Following your suggestions, we have proofread the manuscript afterward and corrected several other typos.

---

### Official Review · Reviewer_k1QU · 2023-10-31

**Soundness:** 3 good
**Presentation:** 1 poor
**Contribution:** 3 good
**Rating:** 6
**Confidence:** 4

**Summary:**

This paper studies knowledge transfer using a SF-DQN architecture in multi-task RL problems (each task is defined by a separate reward function, while they share the same environment). A number of theoretical results are presented for various cases of convergence and knowledge transfer.

**Strengths:**

Theoretical analysis of SF when combined with function approximation is quite interesting and can be insightful.

**Weaknesses:**

- The first half of the paper is confusing as the authors seem to conflate optimal value/policy at the task level with some generic $Q^*$ and $\pi^*$. It was hard to follow and figure out what they mean by the notations and/or possible notational mistake.

- As an important point, note that the GPI theorem induces improvement for the case of several policies but **the same** reward (i.e., only one task). If there are multiple rewards, then maximum of their corresponding value functions is simply meaningless and does not represent any physical concept regarding a new reward function (see the comments bellow). [This is OK when using SF as the $w_i$ corresponds to the current task and only $\psi_j$ comes from some previous task.]

- There were a number of other typos (a proof read aside from the points mentioned above is strongly suggested).

**Questions:**

- If you have multiple tasks, then what do you mean by $Q^*$ and $\pi^*$? In fact, both $\pi^*$ and $Q^*$ are not well-defined unless the reference for optimality is specified. For instance, you may define a total reward as the summation of all the task-level rewards, whose $\pi^*$ and $Q^*$ would be different from the case where a total reward is a *weighted* sum of task-based rewards or some non-linear function of them.

- Eq (13) is not a policy improvement operator (PI is to take an argmax of the current value to induce a new policy).

- Last three lines of page 4 --> First, your reference to $Q^*$ is senseless. More importantly, this setup has nothing to do with your settings. Maximum of several $Q$ functions corresponding to various rewards may not even be a valid $Q$ function for a different reward signal. One can easily construct an MDP with say three tasks and two actions, where in a certain state, $a_1$ is the optimal action for task 3, while $\max_{i\in \{ 1,2 \} } Q_{i}$ give you action $a_2$. Indeed it is theoretically possible that $\max_{i\in \{1,2\}} Q_{i}$ gives you suboptimal action in *all* states.

- Table 1 --> what is $\phi_{i}(\Theta^{*}_{i})$ ? $\phi$ is supposed to be the feature vector.

- Assumption 3 is unclear. Note that $\pi(a|s) \in [0,1]$, Hence, for sufficiently large $C$ this assumption is always true! Am I missing something?

- Is (20) a s fair assumption? For most cases, $K$ would be a large number, so it looks like that the initial approximation of $\psi_1$ must be already very good! Perhaps a better presentation could be that there has to be some pre-training for $\psi_1$ before using this algorithm?

---

> ### Author Response · Authors · 2023-11-22
> **Response to Reviewer k1QU**
>
> We thank the reviewer's comprehensive reading of this paper. The point-to-point responses are posted based on the reviewer’s questions.
>
> *(**Q1**.) The first half of the paper is confusing as the authors seem to conflate optimal value/policy at the task level with some generic $Q^\star$ and $\pi^\star$. It was hard to follow and figure out what they meant by the notations and/or possible notational mistakes.*
>
> (**A1**.) Thank you for your valuable suggestions. We have included a brief introduction to our problem formulation before introducing the preliminaries of $Q$-learning. Specifically, we add a paragraph at the beginning of Section 2 as “in this paper, we address the learning problem involving multiple tasks $\\{ T_i \\}_{i=1}^n$ and aim to find the optimal policy $\pi_i^\star$ for each task $\mathcal{T}_i$. We begin by presenting the preliminaries for a single task and then elaborate on our algorithm for learning with multiple tasks in the following section.” All notations have been modified to follow the same conventions as the subsequent sections to avoid confusion.
>
> Specifically, we include this table to clarify some of the important notations in this paper.
> | Notation | Description |
> | ----------- | ----------- |
> | $Q_i^\star$ | The optimal Q-function for the $i$-th task. |
> | $\pi^\star_i$ | The optimal policy for the $i$-th task. |
> | $\phi$ | Universal transition features. |
> |$w_i$|the reward mapping for $i$-th task.|
> |$\psi^\star_i$|The optimal successor feature for the $i$-th task.|
> |$\psi_i(\Theta_i)$|The function parametered by $\Theta_i$ for learning the successor feature of the $i$-th task.|
> |$\Theta_i^\star$|The ground truth for the successor feature, namely, $\psi_i(\Theta_i^\star)=\psi_i^\star.$|
> |  |  |
>
> *(**Q2**.) As an important point, note that the GPI theorem induces improvement for the case of several policies but the same reward (i.e., only one task). If there are multiple rewards, then maximum of their corresponding value functions is simply meaningless and does not represent any physical concept regarding a new reward function (see the comments bellow). [This is OK when using SF as the $w_i$ corresponds to the current task and only $\phi_j$ comes from some previous task.]  Last three lines of page 4 --> First, your reference to $Q^\star$ is senseless. More importantly, this setup has nothing to do with your settings. Maximum of several $Q$ functions corresponding to various rewards may not even be a valid $Q$ function for a different reward signal. One can easily construct an MDP with say three tasks and two actions, where in a certain state, $a_1$ is the optimal action for task 3, while $\max_{i=1,2}Q_i$ give you action $a_2$. Indeed, it is theoretically possible that $\max_{i=1,2}Q_i$ gives you suboptimal action in all states.*
>
>
> (**A2**.) We appreciate your valuable suggestions, and we apologize for not clearly describing our motivations for using GPI in our settings.
>
> Firstly, as you mentioned, the GPI theorem brings about improvements in cases where multiple policies share the same reward. In our setting, the learning process of the optimal $Q$-function is linked with learning the optimal successor features. When learning these successor features $\psi_i$, we treat $\phi$ as the reward. Given that $\phi$ remains consistent across all tasks, applying GPI across the policies learned from different tasks becomes a viable strategy in learning the new successor feature, thereby enhancing the overall performance in learning the new Q-function. Motivated by the success of GPI, we have introduced the GPI operator within the framework of successor features.
>
> Additionally, existing work (Barreto et al., 2017) empirically verified the efficiency of GPI in the context of successor features and deep reinforcement learning. Furthermore, we have provided theoretical justification (Theorem 2) for the efficiency of GPI in our proposed algorithm SF-DQN. Regarding your examples, if the reward is arbitrarily designed, we agree with you that $\max_{i=1,2}Q_i$ may yield suboptimal actions in all states. However, since the rewards across different policies are connected with $\phi$, $\max_{i=1,2}Q_i$ still performs well in our setting.
>
> Finally, to enhance the clarity of our motivation for incorporating GPI, we have revised the last three lines of page 4.
>
> “When the reward is consistent across different policies, such as ${\pi_i}{i=1}^n$, and considering that the optimal Q-function represents the maximum across the entire policy space, the maximum of multiple Q-functions corresponding to different policies, $\max{1\le i \le n} Q^{\pi_n}$, is expected to be closer to $Q^\star$ than any individual Q-function, $Q^{\pi_i}$. In this paper, the parameter $\phi$ in learning the successor feature is analogous to the reward in learning the Q-function. Since $\phi$ remains constant across different tasks, this analogy has motivated the incorporation of GPI in our setting, even when the rewards change.”

---

> > ### Comment · Reviewer_k1QU · 2023-11-22
> >
> > Thanks for the reply.
> >
> > I don't think I am onboard with your argument. Even though the feature vector $\phi$ is shared among various rewards, the rewards themselves can be completely independent (even in conflict). A simple example is when rewards are identifiers of the feature vector. Say $\phi$ is a vector of 10 components, and $w_{1} = [1, 0, 0, ..., 0]$ and $w_{2} = [0, 0, 1, 0, 0, ..., 0]$. These weight vectors can easily induce two completely orthogonal rewards (not connected even though they share the same $\phi$). This would formally happen if the first and the third components of $\phi$ are independent at a given time. The core problem is that in practice such rewards are quite favourable simply because given a good set of features, these rewards are quite informative (hence, practically useful), but still linear (hence, SF can apply). So, I tend to disagree with your statement. [Further, as you seem to agree, the example I discussed in my original comments also holds.]

---

> ### Author Response · Authors · 2023-11-22
> **Response to Reviewer k1QU (Part II)**
>
> *(**Q3**.) If you have multiple tasks, then what do you mean by $Q^\star$ and $\pi^\star$? In fact, both $\pi^\star$  and $Q^\star$ are not well-defined unless the reference for optimality is specified. For instance, you may define a total reward as the summation of all the task-level rewards, whose $\pi^\star$ and $Q^\star$ would be different from the case where a total reward is a weighted sum of task-based rewards or some non-linear function of them.*
>
> (**A3**.) We apologize for any confusion arising from the notations, which we suspect may be partially caused by the inappropriate use of the term 'multi-task learning' in the abstract. When employing the phrase 'multi-task learning,' our goal is to execute Q-learning for multiple tasks. We later realize that ‘multi-task learning’ is not a proper word because the conventional understanding of 'multi-task learning' is to learn a single policy for multiple tasks. In this paper, instead of learning a single Q-function or policy for all tasks, we will learn a separate $Q$-function for each individual task, which is closer to a continual learning setup. Our algorithm aims to provide a quick adaptation from one task to another new task utilizing the learned knowledge from previous tasks (transfer reinforcement learning).
>
> For the $i$-th sub-task, we use $Q_i^{\star}$ to denote the optimal Q-function for $i$-th subtask.  We use $\pi_i^{\star}$ to denote the optimal policy for $i$-th sub-task. For the $i$-th sub-task, our algorithm returns a Q-function, denoted as $Q_i$.
>
>
>
>
> *(**Q4**.) Eq (13) is not a policy improvement operator (PI is to take an argmax of the current value to induce a new policy).*
>
> (**A4**.) Thank you for pointing out our improper definition, we have updated (13) as
>
> $$\pi(a|s) = \arg\max_{a} \max_\{1\le i \le n+1\} \hat{\psi}_i(s,a)^\top {w}\_{n+1}^\star.$$
>
>
> *(**Q5**.) Table 1 → what is $\phi(\Theta^\star_i)$? $\phi$ is supposed to be the feature vector.*
>
> (**A5**.) We apologize for the typo. $\phi(\Theta^\star_i)$ should be $\psi(\Theta^\star_i)$ in Table 1. $\psi(\Theta^\star_i)$ represents the optimal success feature for the $i$-th task.
>
> *(**Q6**.) Assumption 3 is unclear. Note that $\pi(a|s)\in[0,1]$, Hence, for sufficiently large
>  this assumption is always true! Am I missing something?*
>
> (**A6**.) Thanks for your reflective questions! $C$ has an implicit upper bound that cannot exceed a value less than $1$. As discussed in our first takeaway (T1) in Section 4.1, we conclude that "the generalization error of the learned Q-function scales on the order of $\frac{||w^{(0)}-w^\star||_2}{1-\gamma - N^{-1/2}-C^\star}$," where $C^\star$ is, in fact, a linear function of $C$. It's evident that $C^\star$ has an upper bound in the order of $1-\gamma-N^{-1/2}$. Otherwise, the estimation bound of the Q-function becomes meaningless. Consequently, $C$ also has an upper bound. We apologize for not clearly explaining the relationship between $C$ and $C^\star$ and for not specifying the upper bound of $C^\star$. For more details, we recommend the reviewer check the [General Response (GR)](https://openreview.net/forum?id=s6bKLlF4Pe&noteId=5sCZCZJLVj).

---

> ### Author Response · Authors · 2023-11-22
> **Response to Reviewer k1QU (Part III)**
>
> *(**Q7**.) Is (20) a fair assumption? For most cases, $K$ would be a large number, so it looks like that the initial approximation of $\phi_1$ must be already very good! Perhaps a better presentation could be that there has to be some pre-training for $\phi_1$ before using this algorithm?*
>
> (**A7**.) Thanks for your insightful suggestion! A presentation of the known pre-trained model for $\phi_1$ is indeed a good suggestion, and we will incorporate it into the revised paper. In addition, it is noteworthy that within the NTK framework for supervised learning using neural networks [Zhu et al. 2019 & Arora et al. 2019], a good convergence point—almost as good as the ground truth—is consistently found near random initializations. We believe this observation underscores the practical validation of equation (20). Most importantly, we would like to further clarify that **our initialization assumption in (20) is the state-of-the-art practice in the theoretical analysis of DQNs.**
>
> Specifically, due to the significant convexity of the DQN learning problem, existing DQN results with non-linear activations require various initialization or objective function landscape assumptions to prove convergence, as seen in (i) two-layer assumptions for characterizable Eluder dimension, (ii) the achievability assumption of global optimal in Besov space framework, and (iii) the assumption of a good initialization near the ground truth in analyzing the convergence [Yang et al., 2020; Cai et al., 2019; Xu&Gu, 2020; Du et al., 2020; Nguyen-Tangetal., 2022]. This paper concentrates on deep neural networks, aligning with the conditions outlined in (ii) and (iii). The initialization assumption, as described in equation (20), shares similarities with (iii) and is less stringent than (ii).
>
> Following your suggestion, we have added a paragraph to deliver a better understanding of (20). Specifically, it says “Note that (20) requires a good initialization. Firstly, it is still a state-of-the-art practice in analyzing Q-learning via deep neural network approximation. Secondly, according to the NTK theory, there always exists some good local minima, which is almost as good as the global minima, near some random initialization. Finally, such a good initialization can also be adapted from some pre-trained models. ”
>
>
> [Zhu et al. 2019] Allen-Zhu, et al. "Learning and generalization in overparameterized neural networks, going beyond two layers." NeurIPS, 2019.
>
> [Arora et al. 2019] Arora, Sanjeev, et al. "Fine-grained analysis of optimization and generalization for overparameterized two-layer neural networks." In ICML, 2019.
>
>
>
> *(**Q8**.) There were a number of other typos (a proofread aside from the points mentioned above is strongly suggested).*
>
> (**A8**.) Thank you for bringing the typos to our attention. We have proofread the paper and corrected several errors.

---

> ### Author Response · Authors · 2023-11-22
>
> We thank the reviewer for the prompt response. Sorry that we misunderstood your comments. We agree with your statement that one intuition why SF can be applied is because given a good set of features, these rewards are quite informative and linear. If the reviewer is seeking clarification on how our theorem elucidates this intuition, we offer a brief explanation here.
>
> **The example presented by the reviewer can be clarified through our theorem, as the utilization of SF may result in a worse sample complexity bound when the dimension of the feature vector increases and the distance of the reward mapping between source and target tasks is large.** If the weight vectors are entirely orthogonal for two tasks, these tasks are deemed irrelevant, demonstrating no improvement when employing GPI. Let's consider the learning approaches of DQN and SF-DQN in the scenarios you provided. On the one hand, the distance between $w_1$ and $w_2$ will be significant, resulting in a large $q^\star$, which indicates no improvement when comparing the theoretical results in Theorem 1 (without GPI) and Theorem 2 (with GPI). On the other hand, in the case of SF-DQN, the dimension of the feature vectors must be increased by ten times to satisfy (8), leading to a large upper bound for sample complexity.
>
> When describing the intuition for understanding GPI in SF, we implicitly assume that after SF, the dimension of $\phi$ will either remain unchanged or, at the very least, not scale with the number of tasks, and that the distance between the reward mappings $w_i$ and $w_j$ is small. As you noted, given a good set of features, these rewards are quite informative (hence, practically useful), but still linear (hence, SF can apply). Incorporating irrelevant tasks into the framework of SF will lead to an increase in the feature dimension and a large distance between $w_1$ and $w_2. According to our theorem, this leads to an enlarged sample complexity and, consequently, worse performance.

---

### Official Review · Reviewer_t8DF · 2023-11-01

**Soundness:** 3 good
**Presentation:** 3 good
**Contribution:** 3 good
**Rating:** 8
**Confidence:** 3

**Summary:**

This paper presents a theoretical study over successor feature (SF) based deep Q learning and multi-task transfer learning using generalized policy improvement (GPI). The manuscript provides several theories showing the convergence properties of SF training using DQN, SF training with GPI and transfer gap with GPI.

**Strengths:**

The targeted problem of theoretical analysis for transfer learning using SF is important as SF has achieved many empirical successes in deep RL.

Experimental results verified the gap in SF training and transfer learning as indicated by the proposed theory.

**Weaknesses:**

The proposed framework requires the knowledge of a ground truth feature for state action tuple such that they can linearly represent the true reward function.

Other questiones:

How is the convergence result presented in this paper connected with the GPI theory presented in the original transfer learning with successor feature paper? In the original paper it is assumed that approximated Q functions are given with a deviation $\epsilon$. How would the theories in this work explain the individual terms in the original GPI theory?

Typo:

Sec. 2, paragraph 2: minimize -> maximize, discount factor not shown in the cumulative formula.

**Questions:**

See weakness part.

---

> ### Author Response · Authors · 2023-11-22
> **Response to Reviewer t8DF**
>
> We thank the reviewer's comprehensive reading of this paper and valuable suggestions. The point-to-point responses will be posted based on the reviewer’s questions.
>
> *(**Q1**.) The proposed framework requires the knowledge of a ground truth feature for state action tuple such that they can linearly represent the true reward function.*
>
> (**A1**.) Thank you very much for pointing out this weakness. In the revised paper, we have acknowledged it as a limitation in the conclusion and explored the possibility of addressing it in future directions. One potential solution is to learn $\phi$ using a deep neural network approximation.
>
>
> *(**Q2**.) How is the convergence result presented in this paper connected with the GPI theory presented in the original transfer learning with successor feature paper? In the original paper it is assumed that approximated Q functions are given with a deviation $\epsilon$. How would the theories in this work explain the individual terms in the original GPI theory?*
>
> (**A2**.) Thank you very much for the insightful question. Up to some constant difference, our convergence results presented in equation (25) can be viewed as the approximation error of the $Q$ functions, denoted as $\epsilon$, in equation (9) (Barreto et al., 2017). As equation (25) is in the order of $1/T$, indicating $\epsilon$ should be in the order of $1/T$ when implemented using deep neural network approximation. As shown in equation (10) of (Barreto et al., 2017), the transfer learning error is in the order of $\min_j |w_i-w_j|_2 + \epsilon$. In our theory, equation (27) in our Theorem 3 reveals that the transfer learning error is the summation of one term that is in the order of $\min_j |w_i-w_j|_2$ and one term that is in the order of $1/T$. Here, the $1/T$ term in equation (27) explains the $\epsilon$ in equation (9) (Barreto et al., 2017).
>
> Inspired by your questions, we include a paragraph to connect our theory with the theory presented in (Barreto et al., 2017) as
>
> “Compared with Theorem 2 in (Barreto et al.,2017), the second term of the upper bound in (27) can be explained as ε in Barreto et al. (2017), which is derived from the approximation error of the Q-function to the optimal Q-functions in the previous tasks.”
>
> *(**Q3**.) Sec. 2, paragraph 2: minimize -> maximize, discount factor not shown in the cumulative formula.*
>
> (**A3**.) Thank you very much for pointing out our typo. We have revised the sentence to be “The aim is to maximize the expected discounted sum of reward as $ \sum\_{t=0}^\infty \gamma^t \cdot r_i( s_t, a_t, s\_{t+1})$. ”

---

### Author Response · Authors · 2023-11-22
**General Response (GR) to the upper bound of $C$ in Assumption 3 (@Reviwer k1QU & @Reviewer c6vz)**

We thank the reviewers for their insightful questions. We would like to highlight that  $C$ has an implicit upper bound. To ensure the meaningfulness of the upper bound in (22), the denominator needs to be greater than $0$. Consequently, $C$ is explicitly bounded by $C \leq\frac{(1-\gamma-c_N)(1-\gamma)\rho_1}{|\mathcal{A}|\cdot R_\max}$. In addition, we would like to clarify that the upper bound of $C$ follows the state-of-the-art practice in the theoretical analysis of Q-learning and is necessary for the convergence analysis.

**Firstly, the upper bound of $C$ is necessary for convergence analysis.** Considering the definition of $C$ in Assumption 3, it implies that the difference between the behavior policy and the optimal policy is bounded. In other words, the fraction of bad tuples in the collected samples is constrained. As pointed out by Reviewer c6vz, it is impossible to demonstrate convergence to the optimal Q-function with collected data being all bad tuples. Therefore, such an assumption of constraining the fraction of bad tuples becomes necessary.

**Secondly, the upper bound of $C$ follows the state-of-the-art practice in the convergence analysis of Q-learning with function approximation.** For example, Zou et al. (2019) studied Q-learning with linear function approximation, introducing assumptions similar to our Assumption 3, which also involves a constant $C$. Zou et al. (2019) assume that this constant $C$ cannot be large, ensuring the negativity definiteness of $A + C\lambda I$. Here, $A$ is a square matrix, and $I$ is the identity matrix. This set of assumptions is summarized as Assumption 2 in (Zou et al. 2019). Notably, (Zou et al., 2019) did not provide an explicit upper bound for $C$ due to the unknown nature of the smallest eigenvalue of $A$. However, if the smallest eigenvalue of $A$ is on the order of $\rho_1$, one can derive a comparable upper bound for $C$ as presented in our manuscript. Such a similar assumption can also be found in [Xu & Gu, 2020], which is summarized in Assumption 5.3.

Finally, to address the concerns of the reviewers and avoid further confusion, we have included a Remark after Theorem 1 to clarify the upper bound of $C$.

[Zou et al., 2019] Shaofeng Zou, Tengyu Xu, and Yingbin Liang. Finite-sample analysis for sarsa with linear function approximation. Advances in neural information processing systems, 32, 2019.

[Xu & Gu, 2020] Pan Xu and Quanquan Gu. A finite-time analysis of q-learning with neural network function approximation. In International Conference on Machine Learning, pp.10555–10565.PMLR,2020.

---

### Meta-Review · Area_Chair_socc · 2023-12-05

**Metareview:**

The paper made an interesting step towards studying transfer using successor features in reinforcement learning. The reviewers had some doubts about the results, which were partially addressed during the rebuttal. I would recommend the authors to revise their paper to further clarify these concerns and submit it to the next venue. I feel that the paper is not ready for publication in ICLR at this point.

**Justification For Why Not Higher Score:**

Too many concerns raised during the review, requiring major edits to the paper.

**Justification For Why Not Lower Score:**

N/a

---

### Decision · Program_Chairs · 2024-01-16

Reject